# Advances in Aptamer-Based Conjugate Recognition Techniques for the Detection of Small Molecules in Food

**DOI:** 10.3390/foods13111749

**Published:** 2024-06-02

**Authors:** Xin Deng, Biao Ma, Yunfei Gong, Jiali Li, Yuxin Zhou, Tianran Xu, Peiying Hao, Kai Sun, Zhiyong Lv, Xiaoping Yu, Mingzhou Zhang

**Affiliations:** 1Zhejiang Provincial Key Laboratory of Biometrology and Inspection & Quarantine, China Jiliang University, Hangzhou 310018, China; 15977306909@163.com (X.D.); 16a0701109@cjlu.edu.cn (B.M.); gongyf@cjlu.edu.cn (Y.G.); haopy@cjlu.edu.cn (P.H.); sunkai@cjlu.edu.cn (K.S.); yxp@cjlu.edu.cn (X.Y.); 2Hangzhou Quickgene Sci-Tech. Co., Ltd., Hangzhou 310018, China; qjc1993@126.com; 3College of Life Science, China Jiliang University, Hangzhou 310018, China; xxisjacky@gmail.com (Y.Z.); xiaotianrantongxue@163.com (T.X.); 4Dept Qual Managemet, Inner Mongolia Yili Grp. Co., Ltd., Hohhot 151100, China; lzy@yili.com

**Keywords:** conjugates, small molecules, sensors, rapid detection, food safety

## Abstract

Small molecules are significant risk factors for causing food safety issues, posing serious threats to human health. Sensitive screening for hazards is beneficial for enhancing public security. However, traditional detection methods are unable to meet the requirements for the field screening of small molecules. Therefore, it is necessary to develop applicable methods with high levels of sensitivity and specificity to identify the small molecules. Aptamers are short−chain nucleic acids that can specifically bind to small molecules. By utilizing aptamers to enhance the performance of recognition technology, it is possible to achieve high selectivity and sensitivity levels when detecting small molecules. There have been several varieties of aptamer target recognition techniques developed to improve the ability to detect small molecules in recent years. This review focuses on the principles of detection platforms, classifies the conjugating methods between small molecules and aptamers, summarizes advancements in aptamer−based conjugate recognition techniques for the detection of small molecules in food, and seeks to provide emerging powerful tools in the field of point−of−care diagnostics.

## 1. Introduction

Small molecules are commonly present in food, and they are vital risk factors for foodborne diseases. There are two main reasons for small molecule contamination. One is human activities, which result in pesticide residues [1,2], veterinary drug residues [3,4,5], and other illegal uses of additives [6,7,8]. The other is naturally occurring contaminants, such as toxins produced by fungal metabolism [9,10,11,12]. Regardless of the type of contaminant, once the residual level exceeds the safe limit, it will lead to serious small molecule contaminant problems, affecting food safety and human health.

In order to ensure food safety, numerous countries and organizations have established standards to limit residues in products and materials. Taking pesticides as an example, in 2019, the European Food Safety Authority (EFSA) reviewed the highest safe residual level of pesticides’ active substances, and their limitation was set at 0.01 mg/kg and 0.02 mg/kg in high−water− and high−oil−content samples, respectively [13]. The U.S. Environmental Protection Agency (EPA) set the upper limit of pesticide residues to 0.3 μg/mL in 2021 [14]. In 2022, the Regulations of the Department of Agriculture of China published national food safety standards, including 112 pesticide limits and 290 maximum residue limits, the minimum limit value of which reached 0.02 mg/kg [15]. All regulations highlighted the urgency of establishing fast and accurate small molecule assays.

So far, various methods have been used to detect small molecules. Large−scale instrument analysis has the characteristics of automation and high efficiency [16,17]. Immunoassays have good specificity and low reagent consumption properties [18,19]. Fluorescence rapid detection has the characteristics of accurate analysis and high−throughput performance [20]. However, these traditional methods are not appropriate for on−site screening and quick detection [21], and have poor anti−interference capabilities in complex matrix detection [22,23], resulting in a relatively high false probability. To achieve effective verification, more specific, sensitive, and stable assays are necessary for practical applications.

Aptamers (Apts) were first proposed by Ellington [24] and Tuerk [25], who screened candidate nucleic acid fragments using the systematic evolution of ligands by exponential enrichment (SELEX) method. Their sensitivity is comparable to antigen–antibody interactions, which exhibit high levels of affinity and specificity [26]. After binding to target substances, this new type of recognition molecule undergoes conformational changes, finishing with molecular interactions. Aptamers have been employed as recognition probes, exploring the conjugation principles of aptamer–small molecule conjugates to enhance their combining efficiency and providing widespread applications in the field of small molecule detection [27]. For example, if we use an aptamer to recognize small molecules in detection methods, it will turn the signal from a color or electrochemical signal in an immunosensor to a nucleic acid signal in an aptasensor, which makes signal detection faster and more convenient. Therefore, compared with other methods that are not based on aptamers, the aptamer−based methods have more advantages.

Here, this review has collected methodological research on small molecule detection in recent years, classifying the fundamentals of aptamer–small molecule conjugates and introducing conjugation techniques. It summarizes the research progress in detection technologies based on aptamer–small molecule conjugates, providing new insights into the development of these technologies.

## 2. Aptamer Conjugation Methods

Aptamers have the ability to bind small molecules specifically. In actuality, many challenges appear in practical scenarios, such as cross−reactions and poor adaptability. To solve these problems, researchers have improved the efficiency of aptamer–small molecule conjugates by adding recognition elements and conjugation mediators. Here, three commonly used conjugating methods are presented.

### 2.1. Biotin–Avidin System (BAS)

The BAS is a biological reaction amplification system, which emerged in the 1970s. Its high affinity makes detection analysis more sensitive. However, for fluorescent aptamer sensors, the folding freedom and binding affinity of aptamers are both greatly reduced, which results in a poor sensing performance. To solve this problem, some nanomaterials and chromogenic reagents are added to the conjugate process. A gold nanoparticle, coated with biotinylated and partially complementary DNA (Bio−cDNA−AuNP) and a Cy3−conjugated streptavidin (Cy3−SA), each of which acts as a recognition molecule, energy acceptor, and energy donor, respectively, is added. In the absence of a target, half of an OTA aptamer hybridizes with Bio−cDNA−AuNP and the inhibition of FRET from Cy3−SA to AuNP occurs. On the contrary, a strong and specific interaction between aptamers and OTA reduces the shielding effect, leading to a dose−dependent fluorescence decrease (Figure 1A) [28]. The same principle is applied to the BAS based on Au/Fe_3_O_4_ and the BAS based on GO; the former’s SA is labeled by alkaline phosphatase (Figure 1B) [29], and the latter’s SA is not labeled (Figure 1C) [30].

The cross−reaction between aptamers and small molecules in conjugation always happens. To improve the efficiency in capturing targets, the Bio−modified H2 histamine Apt was incubated with histamine MBs at room temperature, and then SA−HRP was added to initiate the reaction. No matrix effect was observed in the competitive detection of magnetic beads, which proved the applicability and compatibility of aptamers for identifying small molecules (Figure 1D) [31]. Even without MBs, the color reaction between single horseradish peroxidase (HRP) and 3,3′, 5,5′−Tetramethylbenzidine (TMB) can effectively improve the detection performance of the BAS (Figure 1E,F) [32,33].

### 2.2. EDC/Sulfo−NHS Chemical Conjugation

Because of the complexities of covalent conjugation and multi−step preparation, the BAS will take more time to conjugate [34]. EDC is a strong crosslinker and is widely used to conjugate molecules, but the intermediates in the coupling process are unstable [35]. Sulfo−NHS is the preservative most frequently used with EDC, as it can prevent the hydrolysis of intermediates, activate the carboxyl or amino groups to form imide bonds, which are on the surface of small molecules, and form more stable covalent conjugates with aptamers, avoiding complex reactions and improving conjugation efficiency [36]. When using capillary electrophoresis with laser−induced fluorescence (CE−LIF), the conjugates are stable in the binding buffer, but potential dissociation may occur during the process. In order to improve the stability of compounds, EDC/Sulfo−NHS was utilized to enable carboxyl groups to form acyl–amide bonds with amino−terminated complementary sequences (AMOs), resulting in a quantum dot (QD) labeled with AMOs (QD−AMOs), which formed a dual−phase structure with an aptamer. In the absence of the target, the QD−AMOs paired with aptamers to form linear Apt−AMO double chains. Upon the addition of the target, specific binding occurred, which enhanced the conjugation efficiency (Figure 2A) [37].

Sulfo−NHS can not only reduce the dissociation of aptamer–small molecule conjugates, but can also fix the aptamer to the surface of the material for easy detection using a triple−helix molecular switch (THMS) and carboxymethyl cellulose (CMC), followed by EDC/NHS covalent immobilized paper. When OTA became involved, the structure−switching of THMS caused the separation of the dual−labeled signal transduction probe (STP), resulting in the formation of the stem−loop structure of STP with strong fluorescence quenching. Then, the released STP was concentrated on the TR512−peptide functionalized paper, which was modified with CMC−EDC/NHS chemistry coupling via a strong covalent bond. The quenched fluorescent signal of the functionalized paper enabled the quantitative detection of OTA (Figure 2B) [38]. To construct a diclofenac (DCF) detection aptasensor, it was covalently immobilized on the surface of carboxylic−acid−functionalized multi−walled carbon nanotubes (f−MWCNTs), and a glassy carbon electrode (GCE) was modified using EDC/NHS chemistry. The f−MWCNTs provided a reliable matrix for aptamer immobilization with a high grafting density, while the aptamer served as a biorecognition probe for DCF (Figure 2C) [39].

### 2.3. Nucleic Acid Hybridization Method (NAH)

EDC/Sulfo−NHS chemical conjugation has the disadvantages of an unstable coupling efficiency and the fact that its coupling process is not easy to monitor [40]. NAH uses the specificity between aptamers and their complementary chains to complete coupling. When the hybridized aptamer dissociates and preferentially binds to the target, this conjugation method is called hybridization dissociation. At this point, the signal−labeled complementary strand is released, completing the entire detection process by monitoring signal changes. An aptamer with a highly specific binding capability and strong quenching was designed. Thiol−containing aptamers formed covalent bonds with AuNPs, which ensured the binding of fluorescein−labeled complementary strands (FAM−ssDNA) to thiolated AuNPs−OTA aptamers. When OTA appeared, FAM−DNA was released and fluorescence signals were generated (Figure 3A) [41]. AuNPs can be replaced by a molecular beacon (MB); with the assistance of a complementary DNA (cDNA) strand, an MB hybridized with the cDNA to form a duplex structure, in which FAM, labeled at its 3′ end, was separated from BHQ1 labeled at its 5′ end, and there was no fluorescence quenching. Upon the addition of AFB1, the MB preferred to bind with AFB1 rather than cDNA, and adapted into a hairpin structure in which FAM and BHQI were in close proximity, allowing fluorescence quenching to occur. The detection of AFB1 can be rapidly achieved through the measurement of fluorescence intensity decline. Without cDNA, the signal response and sensitivity of the MB probe are weak (Figure 3B) [42].

The conjugation method imposes requirements on the length of a single strand. Too long or too short a length may affect the efficiency of hybridization. To ensure better conjugation efficiency, a novel colorimetric agent named Power Vision (PV) was prepared by selecting an appropriate−length sequence for chloramphenicol (CAP) aptamers. The CAP aptamer, immobilized on magnetic nanoparticles, served as a capture aptamer for CAP (AuMNPs−Apt), while cDNA and PV were labeled on the other set of AuNPs. As the initiator of signal generation, this structure first hybridized with the aptamer and cDNA, which formed the AuMNP−Apt/cDNA−AuNP−PV conjugation. When the CAP was presented, the aptamer bound with it, causing the partial dissociation of cDNA−AuNPs−PV from the conjugate, which was quantitatively analyzed using UV–visible spectroscopy (Figure 3C) [43].

## 3. Aptamer Conjugate Recognition Techniques

### 3.1. Aptamer−Based Up−Conversion Luminescent Nanotechnology (Apt−ULT)

Up−conversion luminescent nanomaterials can transform near−infrared excitation light into ultraviolet or visible light [44]. Additionally, they have high sensitivity levels, a high signal−to−noise ratio, and strong resistance to background interference [45], and these features make them ideal fluorescence donors for ULT. Furthermore, energy receptors are needed to induce fluorescence resonance energy transfer, including nanosheets, graphene, nanoparticles, and carbon dots. The suitable donor and acceptor pairs enable ULT to be widely used in biodetection [46] and food safety [47]. Using near−infrared up−conversion nanoparticles (UCNPs) with OTA aptamers and magnetic nanoparticles (MNPs) modified by complementary oligonucleotides, MNP−UCNP nanocomposites are obtained. When targets exist, the polymer structure opens, and the aptamer binds with it preferentially (Figure 4A) [48]. The linear range of the method is 0.01~100 ng/mL, and the limit of detection (LOD) is 5 pg/mL. It is 10^1^~10^5^ times more sensitive than chromatography or immunoassays (Table 1). In these studies, aptamers are easier to obtain than regular antibodies, and, compared with fluorescent nanoparticles and quantum dots (QDs), they have excellent light stability and luminous intensity, which is fit for the detection of OTA [49]. Magnetic separation−assisted recognition is likely to become a potential tool for the multi−channel ultrasensitive detection of small molecules.

Because of their excellent light and chemical stability, UCNPs have become the competitive candidate donor for luminescence resonance energy transfer (LRET) [54]. LRET−based assays can be performed without repeated separations and pre−processing. Varieties of UCNP−based LRET sensors have been used to analyze targets with different energy receptors [55]. As a type of receptor, MNO_2_ nanosheets were designed for an LRET aptamer sensor (Figure 4B) [56]. This overlap led to luminescence quenching between MNO_2_’s absorption spectrum and UCNPs’ fluorescence emission. When carbendazim (CBZ) was used, it tended to bind with the aptamer preferentially, causing the UCNPs−Apt to fall off the manganese dioxide nanoparticle, which was suitable for detecting CBZ in apple, cucumber, and matcha powder. This sensor showed high levels of sensitivity and specificity, as its LOD was 0.05 μg/mL; comparisons between these methods are shown in Table 2. The accuracy and precision of this method was validated using HPLC, and there was no significant difference. It reduced the fluorescence effects of another eight common pesticides simultaneously, and has good selectivity for CBZ pesticides. In the face of suboptimal synthesis when forming UCNPs, the uniform dispersion effect can be achieved by optimizing the synthesized conditions, further improving the performance of UCNPs.

Compared with QDs, organic dyes, and other luminous materials, UCNPs have a bigger surface area ratio, which tends to result in a stronger surface quenching effect, weakening the fluorescence of the output signal [61]. To solve this problem, core/shell conversion nanoparticles (CS−UCNPs) were prepared as fluorescent donors and graphene oxide (GO) was used as a receptor with LRET aptamer sensors for OTA detection in wine (Figure 4C) [62]. The results showed high luminous efficiency and a high signal−to−noise ratio; compared to the latest detection methods based on nanomaterials reported in recent years, the sensitivity is comparable or even more sensitive (Table 3).

**Figure 4 foods-13-01749-f004:**
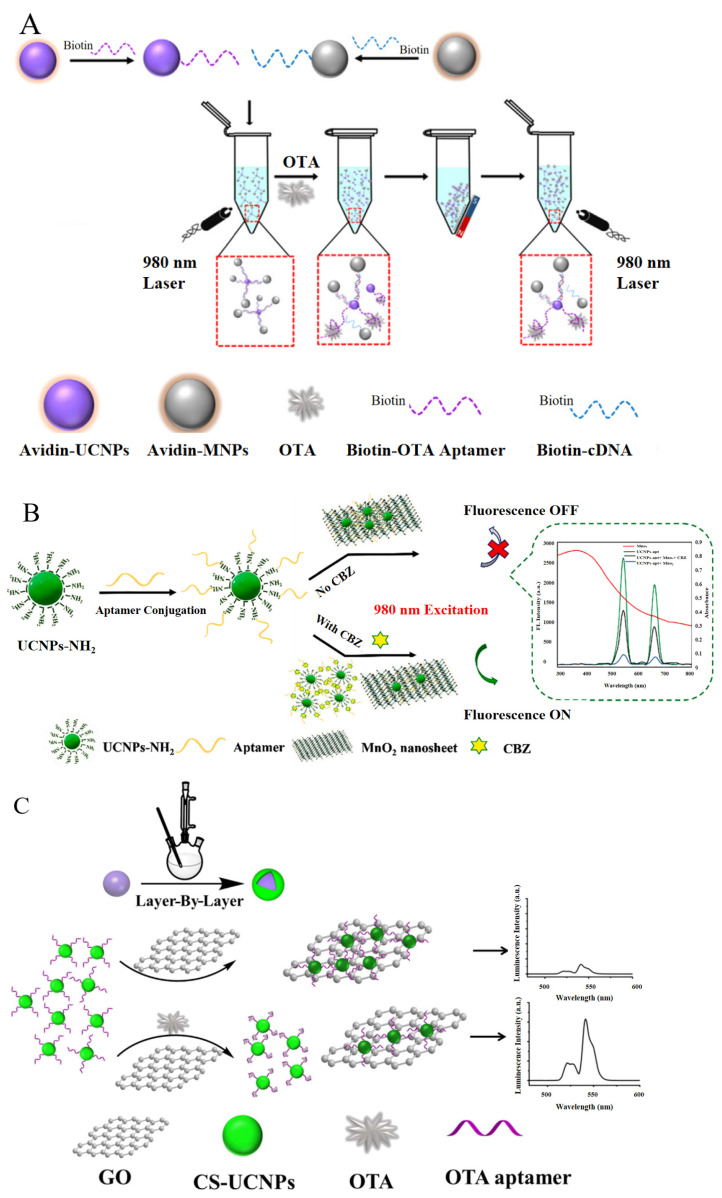
The principles of aptamer−based up−conversion luminescent nanotechnology. (**A**) The detection of ochratoxin A in beer. Reproduced with permission from [48]; published by Elsevier, 2016. (**B**) The detection of carbendazim in apples, cucumbers, and matcha powder. Reproduced with permission from [56]; published by Elsevier, 2021. (**C**) The detection of ochratoxin A in wine. Reproduced with permission from [62]; published by Elsevier, 2017.

### 3.2. Aptamer−Based Immunoassay Methods

#### 3.2.1. Aptamer−Based Enzyme−Linked Immunity Assay (Apt−ELISA)

In a traditional ELISA, a primary antibody and an enzyme−labeled secondary antibody are selective for small molecules, and display the results through a color reaction [67]. However, antibodies are susceptible to chemical treatment, which can result in the loss of epitope and unstable properties [68,69]. Aptamers have strong stability, and they can combine with other nanomaterials through modification and immobilization to significantly improve detection efficiency [70]. As an alternative material for antibodies, aptamers not only show excellent repeatability but also simplify cumbersome operations [71]. By contrast, traditional enzymes are not easy to preserve and are unstable. Additionally, nano enzymes have high catalytic efficiency. As a kind of simulated enzyme with catalytic activity and unique functions, they can be prepared on a large scale as excellent substitutes for labeling enzymes [72].

AuNPs–nano enzymes can be obtained through the combination of AuNPs and nano enzymes. For their stability and affordability, AuNPs–nano enzymes are widely used. They were labeled on the aptamer, and the complex serves as a substitute for secondary antibodies in ELISA. In a direct competition ELISA, the AuNPs–nano enzymes−labeled aptamer was used as an enzyme−labeled antibody for ampicillin residue testing in commercially available milk (Figure 5A) [73]. Compared with methods based on metal–organic framework materials (MOFs) [74] and molecularly imprinted polymers (MIPs) [75], the linear range of this method is wider and the detection limit is 10^3^ times lower. This Apt−ELISA platform was the first reported for ampicillin detection, with high levels of specificity and repeatability; however, the DNA−functionalized AuNP probe used in another study was prepared by the self−assembly of thiolated oligonucleotides (SH−DNA) on the surface of nanoparticles [76]. It is worth noting that the SH−DNA chains exhibit a coiled and folded conformation, which means that the spatial hindrance of DNA and its complementary chains increases [77], affecting the detection performance of Apt−LFD. In addition, it can also be used to detect other small molecules in food, such as ochratoxin A (Figure 5B) [78], tetracycline (Figure 5C,D) [79,80], and aflatoxin B1 (Figure 5E,F) [81,82] (Table 4).

#### 3.2.2. Aptamer−Based Lateral Flow Chromatography Strip (Apt−LFD)

The popular lateral flow dipstick (LFD) has a large coefficient of variation between batches and a wide error margin. These issues cause inconvenience and are a barrier to the widespread application of LFDs [83]. Considering its admirable flexibility, an Apt can replace traditional antibodies to bind small molecules successfully, which may only recognize biological molecules [84]. An aptamer−mediated transversal flow test strip based on nano enzymes has been developed, which can provide results that are visible to the naked eye, and it has been made available for the on−site quantitative analysis of acetamiprid (Figure 6A) [85] and AFB1 (Figure 6B) [86], using smartphones.

It has been reported that poly adenine (polyA) DNA fragments can serve as anchor blocks and preferentially bind to the surface of AuNPs [87]. Additionally, due to the fact that polyA−DNA can appear in extended and upright conformations on AuNPs, it improves the speed and efficiency of hybridization [88]. There was a method developed that was based on AuNPs@polyA−cDNA Apt−LFDs with nanoprobes, which solved the problem of spatial hindrance caused by hybridization and had lower production costs. It was successfully applied to the detection of acetamiprid in tomatoes (Figure 6C) [89] and kanamycin in honey (Figure 6D) [90], the limits of detection were 0.33 ng/mL and 250 ng/mL, respectively. Compared with some fluorescent methods [91] and typical colorimetric methods [92] for acetamiprid, this technology was comparable in sensitivity, and it was much more convenient to use than other methods. Up to now, an Apt−LFD has been proven to detect pesticides, biotoxins, and antibiotics, including microcystin−LR (Figure 6E) [93] and zearalenone (Figure 6F) [94]. However, the sensitivity of aptamer−based immunoassay methods is limited, and their detection performance needs to be improved (Table 5).

**Figure 6 foods-13-01749-f006:**
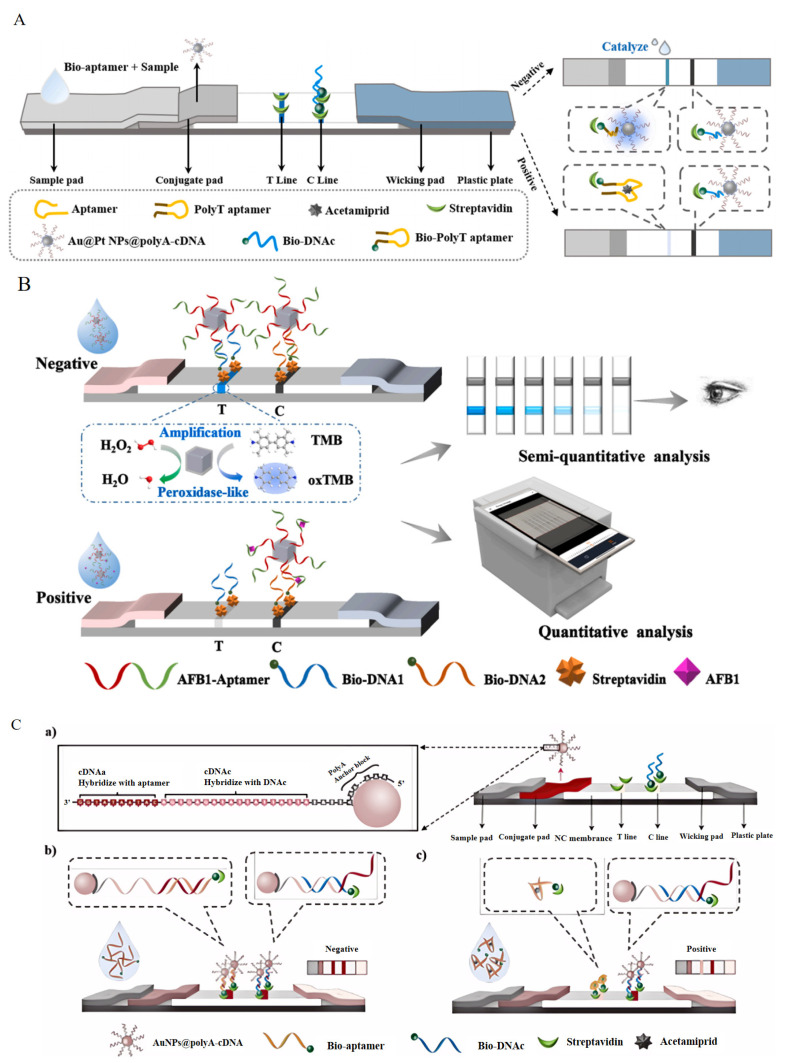
The principle of aptamer−based lateral flow chromatography strip. (**A**) The detection of acetamiprid using a nano enzyme–Apt LFD. Reproduced with permission from [85]; published by Elsevier, 2023. (**B**) The detection of aflatoxin B1 using a nano enzyme–Apt LFD. Reproduced with permission from [86]; published by Elsevier, 2024. (**C**) The detection of acetamiprid using a AuNPs−Apt LFD. Reproduced with permission from [89]; published by Elsevier, 2022. (**D**) The detection of kanamycin using a AuNPs−Apt LFD. Reproduced with permission from [90]; published by Springer, 2022. (**E**) The detection of microcystins using a AuNPs−Apt LFD. Reproduced with permission from [93]; published by Elsevier, 2023. (**F**) The detection of zearalenone using a AuNPs−Apt LFD. Reproduced with permission from [94]; published by AMER CHEMICAL SOC, 2018.

### 3.3. Bio−Aptasensors

#### 3.3.1. Fluorescent Aptasensors (FASs)

FASs are classified into labeled and unlabeled categories [95]. Biological molecules and aptamers in the natural state rarely have self−luminous properties; therefore, it is necessary to design a FAS with the ability for simple, feasible, and rapid detection [96]. The most commonly used fluorescent label is carboxyfluorescein (FAM) [97]. In order to utilize the spectral characteristics of UCNPs and the adsorption capacity of metal–organic frameworks (MOFs) in detecting enrofloxacin (ENR), a super−sensitive and highly selective fluorescence sensor was developed (Figure 7A) [98]. The modified UCNP utilized FRET as the donor and MOFs−MIL−101 (CR) as the receptor. In the presence of ENR, the aptamer was prioritized to bind and its structure was changed, thus reducing the influence of bases on the receptor surface and fluorescence recovery. Compared with other fluorescence sensors [99] and electrochemical methods [100] based on other materials, this method has significant advantages in sensitivity levels and analysis range. The detection limit is 34 pg/mL, and the applicability has been verified using real shrimp samples, demonstrating it can be applied to the detection of ENR in seafood.

To better promote practical detection and on−site screening, it is necessary for a FAS to improve its sensitivity and efficiency. Paper is cheap and easy to preserve, with good biocompatibility and biodegradability [101]. These excellent peculiarities make it the best substrate for sensors. A paper−based FAS used for the simultaneous detection of zearalenone (ZEN) and OTA was established. Targets were migrated from the sample area to the detection area through dual channels, and then produced green and blue fluorescence, respectively. This could be observed visually and the output detection results were available on a smartphone (Figure 7B) [102]. The detection limits of this method were 0.44 ng/mL and 0.098 ng/mL, respectively. Its sensitivity was not as high as that of a multi−polymer−based FAS [98], but it was better than that of traditional fluorescent microsphere test strips [103]. It had the characteristics of visualization, greenness, and wide adaptability, which was suitable for the on−site multiple detection of small molecules.

Although aptamer sensors with fluorescent labels enable convenient detection, the design procedure is time−consuming and costly [104]. Therefore, researchers have turned their attention to unlabeled sensors, which are low−cost, easy to operate, and have detection limits that are comparable to labeled aptamer sensors, as shown in Table 6. They also obtain good results when detecting small molecules in foods, such as pesticide residues in rice (Figure 7C) [105] and aflatoxin B1 in soy sauce (Figure 7D) [106].

**Figure 7 foods-13-01749-f007:**
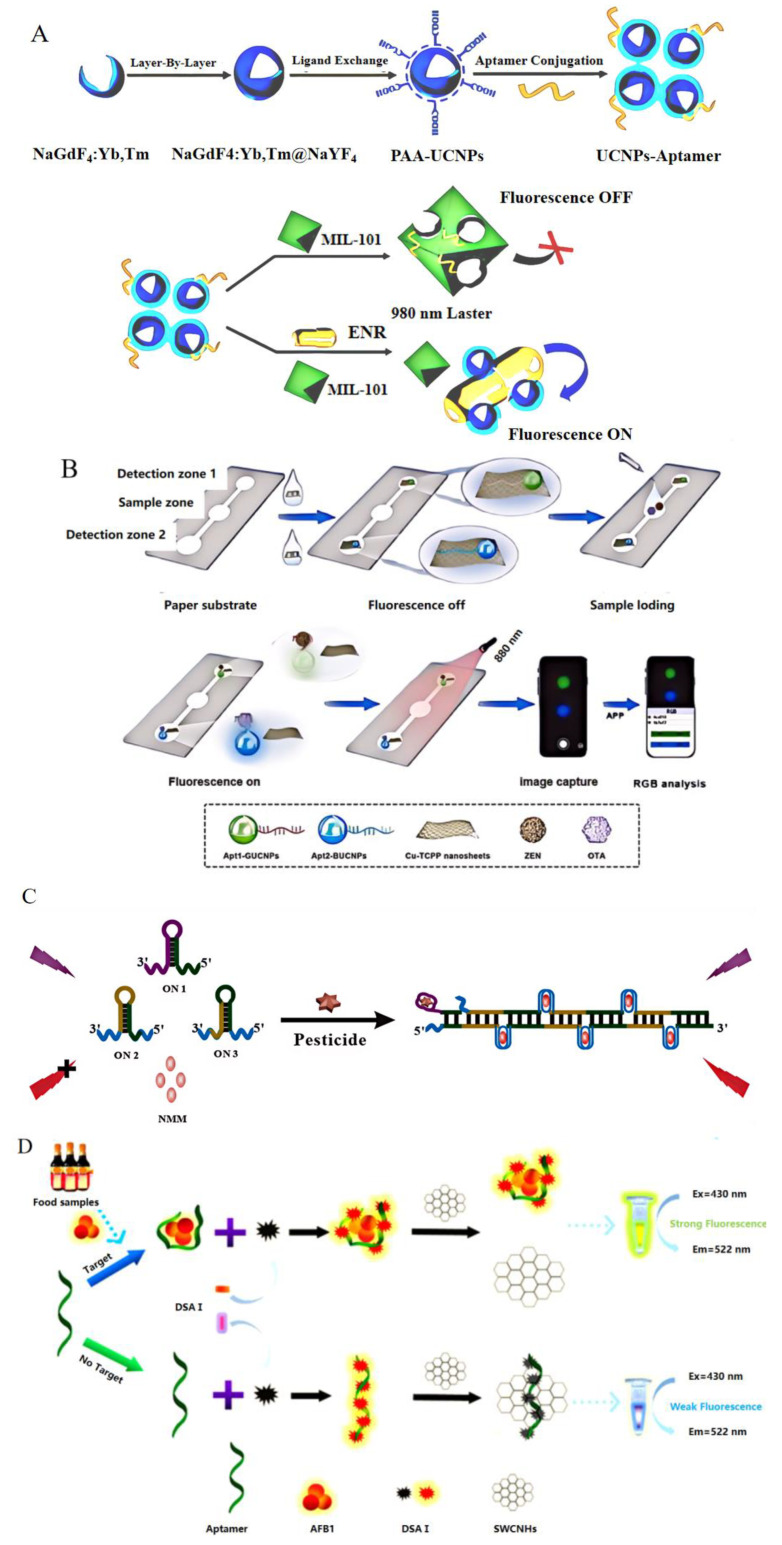
The principles of fluorescent aptasensors. (**A**) The detection of enrofloxacin using a labeled FAS. Reproduced with permission from [98]; published by Elsevier, 2022. (**B**) The detection of zearalenone and ochratoxin A using a labeled FAS. Reproduced with permission from [102]; published by Elsevier, 2023. (**C**) The detection of pesticides using a label−free FAS. Reproduced with permission from [105]; published by Elsevier, 2023. (**D**) The detection of aflatoxin B1 using a label−free FAS. Reproduced from [106].

#### 3.3.2. Electrochemical Aptasensor (EAS)

An EAS is a new type of biosensor developed in recent years, wherein the aptamer, fixed on the electrode surface, will not trigger signal reactions in the absence of a target. When the target appears and binds to the aptamer, the structure changes, bringing measurable variation to the sensor [107]. When OTA is present, the configuration of the hairpin DNA (h−DNA) changes and opens, which consists of an aptamer and its complementary strand DNA (cDNA), and the cDNA that supplies the reporter DNA (rDNA) is exposed. After the polymer–nanoparticle composites (POPD−GNSs) with rDNA are led to the electrode surface, electrical signals are generated (Figure 8A) [108]. The practicability and reliability of this method were verified by detecting OTA in coffee; a low limit of detection of 1 pg/mL was obtained, and the monitoring range was from 2 pg/mL to 1 ng/mL, but the limitations of the electrode led to a decrease in conductivity. AuNPs have good conductivity and a specific surface area [109]. When combined with the aptamer, they can greatly improve the performance of an EAS; when it was successfully applied for the detection of melamine (MEL) residue in milk, its selectivity, reproducibility, and stability were significantly improved, with a LOD of 6.7 × 10^−13^ M [110]. Compared with previous reports on melamine detection and analysis methods, this method has a relatively wide linear range and a lower detection limit [111,112]. This means that the selectivity and sensitivity levels have been greatly improved.

From single AuNPs, graphene aerogel (GA), and graphene quantum dots (GQDs), AuNPs−GQD and AuNPs−GQD−GA hybrids were fabricated and used for the fabrication of electrochemical sensors (ESs), which could achieve the requirements of many aspects of ESs [113,114]. However, the present EAS still requires an additional redox probe to produce electrochemical signals for the detection of non−electroactive analytes, because it does not possess the functions of a redox probe (Figure 8B) [115]. There is an EAS that uses Ferrocene (Fc) as a single redox reporter to achieve the self−calibrating electrochemical detection of AFB1 (Figure 8C) [116]. When compared to other developed aptasensors, its accuracy and sensitivity were much better than those of a luminescence resonance energy transfer aptasensor (LERT) [117], but these were equal to those of the other reported EASs [118], with a linear range of 0.1~10,000 pg/mL and a LOD of 0.012 pg/mL. There has been some research undertaken on using single Fc (Figure 8D) [119] and other redox reporters to detect small molecules in food, such as MBs (Figure 8E) [120], [Fe (CN)6]^3−/4−^ (Figure 8F) [121], MB−Fc (Figure 8G) [122], and THI−Fc (Figure 8H) [123].

AFB1 and OTA are incredibly relevant to food safety, as they are highly likely to simultaneously occur in food samples. Using a method involving two redox probes, hemin@HKUST−1 and ferrocene@HKUST−1, which could achieve simultaneous detection of these two mycotoxins (Figure 8I) [124], shortened the assay time and reduced the cost, and it was able to test very low concentrations of the targets in cornflour samples. However, each EAS can only be used in one experiment, as it is disposable. The EAS methods with different labels/probes are shown in Table 7.

**Figure 8 foods-13-01749-f008:**
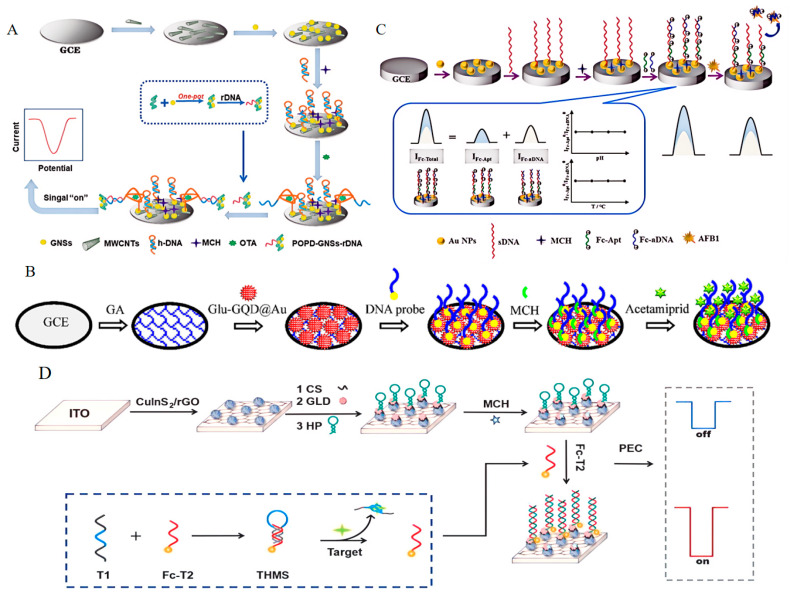
The principles of electrochemical aptasensors, based on different redox probes. (**A**) The detection of ochratoxin A using an EAS. Reproduced with permission from [108]; published by ROYAL SOC CHEMISTRY, 2022. (**B**) The detection of acetamiprid using an EAS based on Glu−GQD/Au. Reproduced with permission from [115]; published by Elsevier, 2019. (**C**) The detection of aflatoxin B1 using an EAS based on Fc. Reproduced with permission from [116]; published by Elsevier, 2022. (**D**) The detection of amoxicillin using an EAS based on Fc. Reproduced ]with permission from [119]; published by Elsevier, 2024. (**E**) The detection of aflatoxin B1 using an EAS based on MB. Reproduced with permission from [120]; published by AMER CHEMICAL SOC, 2023. (**F**) The detection of bisphenol A using an EAS based on [Fe (CN)6]^3−/4−^ Reproduced with permission from [121]; published by Elsevier, 2019. (**G**) The detection of patulin using an EAS based on MB−Fc. Reproduced from [122]. (**H**) The detection of malathion and chlorpyrifos using an EAS based on THI−Fc. Reproduced with permission from [123]; published by Elsevier, 2022. (**I**) The simultaneous detection of aflatoxin B1 and ochratoxin A using an EAS based on hemin@HKUST−1 and Fc@HKUST−1. Reproduced with permission from [124]; published by Elsevier, 2024.

#### 3.3.3. Colorimetric Aptasensors (CASs)

The use of CASs enables the researcher to directly read the results using the naked eye, without complex and expensive instruments, which enables broad application prospects in field diagnosis [125]. Because of surface functionalization, AuNPs have been widely used in CASs due to their excellent optical properties. AuNPs have been used to link capture sequences (SH−poly A−cDNA), and a AuNPs@SH−poly A−cDNA nanoprobe has been designed. The test strip, based on the use of AuNPs@SH−poly A−cDNA as a competitive aptamer, was developed to detect AFB1. In the absence of AFB1, the aptamer bound to the nanoprobe, and the cDNA was prevented from binding to the complementary fragment of the test line (DNA_T_). The AFB1 preferentially bound to the aptamer, and the free nanoprobes bound to DNA_T_, generating signals (Figure 9A) [126], making the test LOD below 10 ng/mL and the visual detection limit 50 ng/mL. There are different CASs for aflatoxin B1 detection in food (Table 8). This method has a better linear range and a lower detection limit than CASs based on different signal probes, such as AuNPs (Figure 9B) and MNPs (Figure 9C), but its sensitivity is not as good as CASs based on labeled Cy5 (Figure 9D).

Multiple rounds of screening are usually required to obtain better aptamers, while traditional screening can only provide a limited number of candidate sequences. AuNPs not only realize the visualization, but also assist in screening highly active Apts from a large number of sequences through color changes. Based on this principle, SELEX was used to select specific aptamers, then qRT−PCR was utilized to detect them and to conduct high−throughput sequencing for an enriched library. A CAS based on AuNPs was used to screen candidate aptamers and identify them initially. With a selection based on the highest affinity, 5′−biotin aptamer 47 was chosen to develop a CAS based on AuNPs for the detection of clenbuterol in pork, the LOD of which was below 0.18 ng/L. This was the first report of a universal strategy including the use of library fixation, Q−PCR monitoring, high−throughput sequencing, and AuNP biosensor identification to select aptamers specific for small molecule detection (Figure 9E) [130]. This method proposes multiple screening strategies, and it provides ideas for the screening and identification of other aptamer–small molecule conjugations.

**Figure 9 foods-13-01749-f009:**
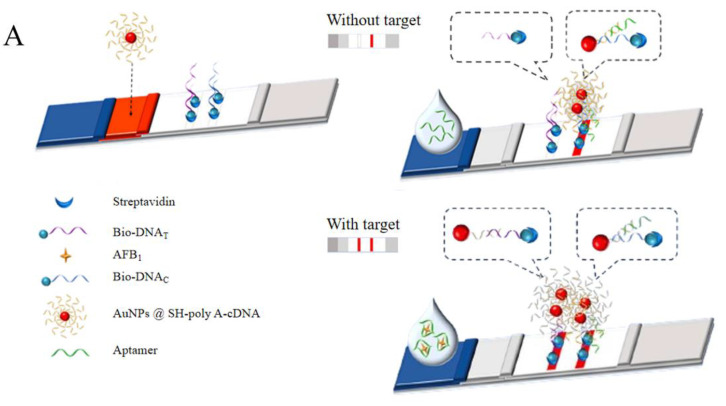
The principles of colorimetric aptasensors based on different signal probes. (**A**) The detection of aflatoxin B1 using a CAS based on AuNPs@SH−poly A. Reproduced with permission from [126]; published by Elsevier, 2023. (**B**) The detection of aflatoxin B1 using a CAS based on AuNPs. Reproduced with permission from [127]; published by Elsevier, 2020. (**C**) The detection of aflatoxin B1 using a CAS based on MNPs. Reproduced with permission from [128]; published by AMER CHEMICAL SOC, 2021. (**D**) The detection of aflatoxin B1 using a CAS based on Cy5. Reproduced with permission from [129]; published by ELSEVIER, 2017. (**E**) The detection of clenbuterol using a CAS based on AuNPs. Reproduced from [130].

#### 3.3.4. Immuno−Aptasensors (IASs)

Immunosensors utilize antigen–antibody interactions for detection, while the working principle of an IAS is its high−specificity binding of nucleic acid aptamers to the target. Although both methods use the specific interactions between biomolecules to detect the target, the latter can detect small molecules that cannot be detected by the former, with good prospects in intelligence, quick responses, and strong adaptability. The most commonly used type of IAS is based on a lateral flow strip (LFS). It is similar to the CAS, which can be observed using the naked eye [131]. The user−friendly format and long−term stability are the advantages of this aptamer sensor [132]. In addition to qualitative analysis, IAS−LFS can also be quantified by scanners [133].

An IAS based on dual probes uses probe 1 fixed on a test line, which competes with the target and binds to the AuNPs−Apt, before the complex is migrated to the coupling pad. If the target is present, it will bind to the aptamer–AuNP conjugate, preventing AuNPs−Apt from interacting with probe 1. If the target is not present, AuNPs−Apt will bind to probe 1. Therefore, the strength of the test line decreases with the concentration of the target. In order to ensure effectiveness and authenticity, whether the target is present in the sample or not, probe 2 will bind to the AuNPs−Apt and display a color on the control line. At present, it has been used to detect ochratoxin A in apple juice and milk (Figure 10A) [134]. When compared to colorimetry [135], visual inspection [136], and fluorescence [133], LFS−IAS had a low visual LOD and quantitative LOD, and the total assay time was 20 min with simple steps. Furthermore, it can also be used to detect aflatoxin B1; the detection limit of an IAS assembled using AuNPs−Apt for aflatoxin B1 in cornflour was 0.1 ng/mL. The sensor had no response to type−G aflatoxin, B2, M1, or zearalenone, which showed it has good selectivity (Figure 10B) [137].

As is known, Cy5 is a label used to mark biomolecules. When it is used for LFS−IAS, the complementary strand competes with the target to bind the Cy5−Apts on the test line. Among them, streptavidin modifies the complementary chains of the aptamers. In the absence of the target, the Cy5−Apts are first combined with a partially complementary strand. Contrarily, when the target is present, the Cy5−Apts preferentially bind to the target. This method can be used for the detection of type−B aflatoxins in nuts and dried figs, with a limit of detection of 0.16 ng/mL (Figure 10C) [138]. It has an equivalent sensitivity to antibody strips and other aptamer strips, but a good linear range. Compared with the label−free immunosensors (LFISs), these IASs have higher levels of sensitivity, but the lower limit of their detection range is smaller than that of LFISs; not only can they detect solid food, but they can also detect liquid samples (Table 9).

**Figure 10 foods-13-01749-f010:**
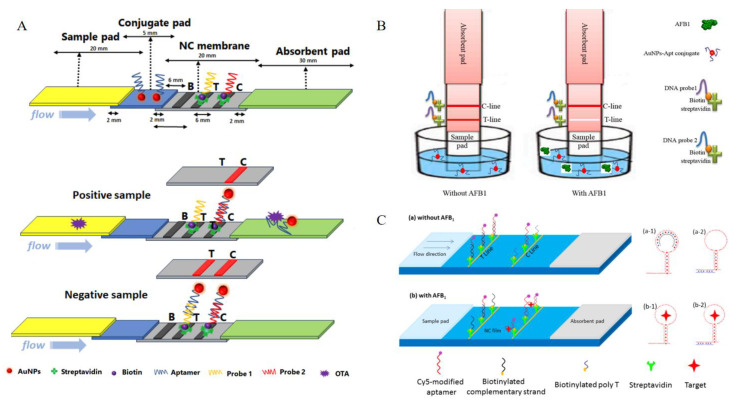
The principles of immuno−aptasensors based on different labels/probes. (**A**) The detection of ochratoxin A using an IAS based on AuNPs. Reproduced from [134]. (**B**) The detection of aflatoxin B1 using an IAS based on AuNPs. Reproduced with permission from [137]; published by Elsevier, 2023. (**C**) The detection of type−B aflatoxins using an IAS based on Cy5. (a) In the absence of AFB1, the Cy5-labeled aptamer was first combined with its complementary strand at the test-line (T-line) (a-1); the remaining aptamer was then combined with the poly T of the control line (C-line) (a-2). (b) In the presence of AFB1, the Cy5-labeled aptamer preferred to bind with AFB1 (b-1), and the fluorescence intensity of the T-line decreased. The aptamer bound to AFB1 was still bound to poly T of the C line(b-2). Reproduced from [138].

### 3.4. Aptamer−Based Nucleic Acid Detection Technology (Apt−NAT)

With the rapid development of molecular biology, the detection sensitivity of macromolecules represented by nucleic acids has increased to the level of single−copy sensitivity. The combination of aptamer−based and nucleic acid−based detection technology expands this dynamic detection range, which is conducive for the efficient and sensitive detection of small molecules. An aptamer−assisted real−time fluorescent quantitative PCR (Apt−qPCR) method, based on the target−induced dissociation (TID) effect, was developed, wherein the OTA aptamer was fixed on a magnetic bead coated with d(T)_25_. The sequence was complementary to the aptamer, which was used as a linker. When OTA appeared, due to the presence of the TID effect, the aptamer was released from the dT beads into the supernatant, and was quantitatively analyzed using a qPCR assay (Figure 11A) [142]. The detection range of OTA in beer was between 0.039 and 1000 ng/mL, and its LOD was as low as 9 μg/mL. Compared to a FAS and an EAS, it had a lower detection limit and a wider detection range [143,144]. This method ultimately amplified the reaction signal through PCR; therefore, compared to many other reported detection methods, this detection method is not fast, as the whole assay time may take 2.5 h.

In addition, Apt−qPCR relies on precise temperature control instruments, so the amplification efficiency of this method would be unstable with temperature fluctuations. Using loop−mediated isothermal amplification (LAMP) instead of a PCR can achieve the goal of rapid and efficient OTA detection. The number of Apt−OTA conjugates reduced on the electrode surface after the target was identified. The remaining Apt−OTA served as internal primers to initiate LAMP. The LOD reached 10 fM, suggesting the noteworthy amplification of the LAMP technique, which had simple steps for sensor preparation and simultaneously demonstrated good stability (Figure 11B) [145]. This method’s sensitivity was higher than other reported aptasensors, such as FASs [146] and UV–vis−based aptasensors [147].

In addition to Apt−LAMP, there are several other aptamer−based isothermal amplification techniques. The competitive colorimetric method based on aptamers was established using a hybrid chain reaction (HCR) to catalyze AuNPs. After hybridization with complementary chains on magnetic beads, the aptamer was combined with saxitoxin (STX) competitively, magnetic separation was performed to obtain the Apt−STX conjugations, which are suitable for subsequent signal transduction, and an HCR was triggered to generate double−stranded DNA (dsDNAs) (Figure 11C) [148]. This study significantly enhanced the catalytic capacity of AuNPs. The LOD was reduced to 42.46 pM, and the linear range was 78.13~2500 pM when detecting shellfish. The sensitivity of this method was better than FASs and EASs [149,150], but its linear detection range and the converted LOD were worse than other methods of the same type [151].

The strand displacement reaction (SDA) is simpler and more stable than other isothermal amplification techniques, and can produce a large amount of single−stranded DNA (ssDNA). An SDA based on a patulin aptamer (Apt−PAT) has been designed, wherein PAT and aptamer complementary chains compete to bind the aptamer, and MNP−Ap−PAT and MNP−Apt−cDNA are obtained through magnetic separation. The cDNA triggers the SDA reaction to produce a large amount of ssDNA. This ssDNA forms the TTAPE−quadruple that binds to the TTAPE dye and amplifies the fluorescent signal. Compared with other methods, the lower LOD was 0.042 pg/mL and it had a wide linear range that applied to the detection of apple juice and grape juice (Figure 11D) [152]. When compared with MIPs−ES, also used for detecting apple juice, the sensitivity was 10^4^ times higher and the linear range was also expanded 50−fold [153]. The detection performance of the Apt−NAT mentioned above is shown in Table 10.

**Figure 11 foods-13-01749-f011:**
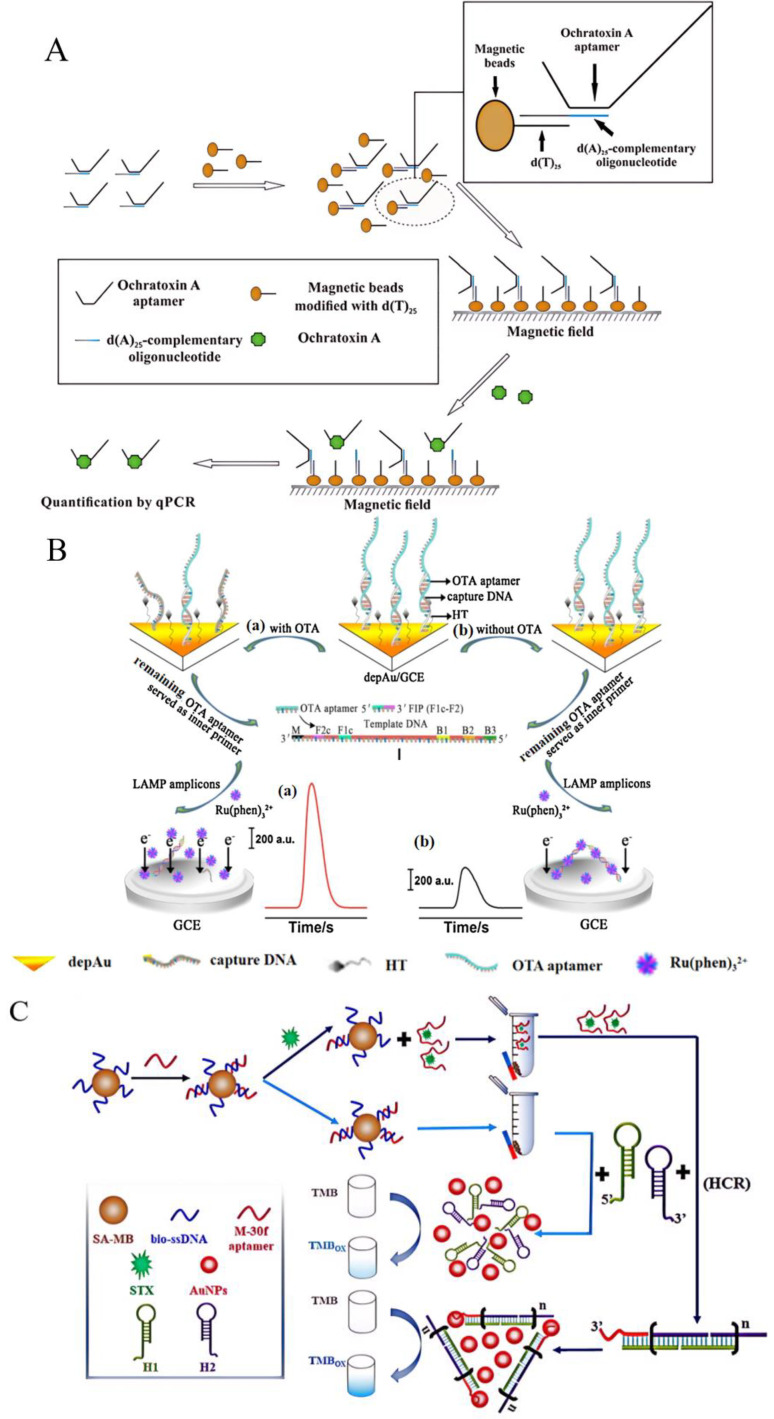
The principle of aptamer−based nucleic acid detection technology. (**A**) The detection of ochratoxin A using Apt−qPCR. Reproduced with permission from [142]; published by WILEY, 2017. (**B**) The detection of ochratoxin A using Apt−LAMP. (a) In the presence of OTA, the OTA aptamer bond with target OTA and subsequently left off the electrode, which effectively decreased the immobilization amount of OTA aptamer on electrode. Then, the remaining OTA aptamers on the electrode served as inner primer to hybridize with template DNA and thus initiated the LAMP reaction. (b) In the absence of OTA, The ECL indictor Ru(phen)_3_^2+^ binding to amplicons caused the reduction of the ECL intensity. Reproduced with permission from [145]; published by ELSEVIER, 2014. (**C**) The detection of saxitoxin using Apt−HCR. Reproduced with permission from [148]; published by ELSEVIER, 2021. (**D**) The detection of patulin toxin using Apt−SDA. Reproduced with permission from [152]; published by ELSEVIER, 2023.

## 4. Conclusions

Small molecules are critical risk factors in food. Once maximum residue limits are exceeded, small molecules will result in hazards to human health and food safety. Therefore, it is necessary to realize effective and sensitive detection techniques for small molecules. Due to the fast speed of food consumption and circulation, the requirements for the detection of small molecules are constantly changing. For great stability and high specificity levels, aptamers maintain high selectivity for small molecules after multiple rounds of screening, and form recognizable aptamer–small molecule conjugates. By using appropriate identification methods, specific aptamer–small molecule conjugates can be detected, thereby setting up a low−cost and highly efficient platform.

There are three aptamer−based conjugate methods introduced in this study, each of which has its own characteristics for aptamer−small molecule conjugation. The BAS has a high affinity and selectivity when conjugating the aptamer and the target, but cumbersome steps are always required to finish the process. EDC/Sulfo−NHS chemical conjugation has a quick conjugation speed, but it is difficult to determine whether cross−linking is successful or not. NAH does not require the addition of additional connecting molecules or chemical reagents in the conjugation system, as it uses the basic principle of nucleic acid complementary pairing to achieve the conjugate. In other reported research, they mostly used simple labels and nanomaterials on the aptamer or its complementary chains to achieve the goal of identifying conjugates.

In the core of this study, four detection methods are mentioned, most of which can reach a lower LOD than the methods that are not based on aptamers, thereby demonstrating the advantages of aptamers for the signal recognition of small molecules. Among these, Apt−ULT relies on the up−conversion of nanoparticles for detection, whose synthesized morphology is not ideal at all times. There are two types of aptamer−based immunoassay, namely Apt−ELISA and Apt−LFD, which utilize nano−enzymes and nanomaterials to improve detection performance. However, the former usually takes a long time to incubate, which makes on−site inspection difficult. The latter shortens the analysis time, but it has a weakness of signal contrast and catalytic color uniformity. There are four types of bio−aptasensors used for detecting small molecules. Both FASs and CASs belong to the category of visual detection, which provides convenience when observing results. Their sensitivity and accuracy are not as good as other bio−aptasensors. In all the summarized reports, an EAS based on different redox probes is the most sensitive bio−aptasensor for small molecule detection, which can reach the level of pM, or even fM. An IAS is better than label−free immunosensors in some aspects, but it is comparable to FASs and CASs in terms of performance. In Apt−NAT, the isothermal method is better than the poikilothermic method, because it can avoid temperature instrument dependence and save analytical time.

In summary, the existence of aptamers has solved the performance shortcomings of some technologies, enabling their successful application in the trace detection of small molecules. These methods have good adaptability and practicality and do not require complex sample pretreatment, achieving the requirements of rapid detection. For biological aptamer sensors in particular, their excellent adaptability and portability will enable them to achieve on−site applications in the near future. Some have developed corresponding kits, test strips, and other convenient tools to demonstrate their broad application prospects in practical production, such as in Apt−ELISAs and Apt−LFDs. These technologies based on aptamers will further leverage their unique advantages to provide more efficient and sensitive solutions to ensure food safety.

## Figures and Tables

**Figure 1 foods-13-01749-f001:**
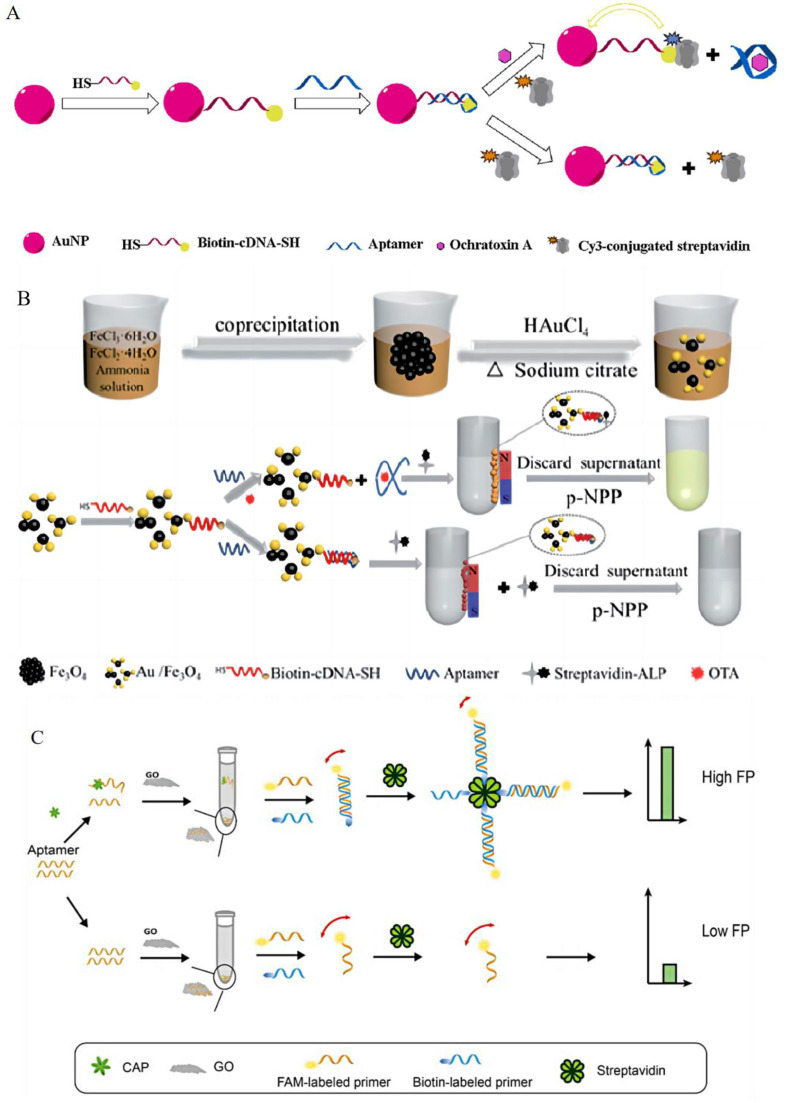
The principles of the biotin–avidin system. (**A**) The detection of ochratoxin A in wheat and green coffee beans. Reproduced with permission from [28]; published by Elsevier, 2016. (**B**) The detection of ochratoxin A in peanut, corn, and wine. Reproduced from [29]. (**C**) The detection of chloramphenicol in honey. Reproduced with permission from [30]; published by Elsevier, 2019. (**D**) The detection of histamine in meat, fish, and beverages. Reproduced with permission from [31]; published by AMER CHEMICAL SOC, 2019. (**E**) The detection of ochratoxin A in corn. Reproduced with permission from [32]; published by AMER CHEMICAL SOC, 2020. (**F**) The detection of fumonisin B1 in beer and corn. Reproduced from [32,33].

**Figure 2 foods-13-01749-f002:**
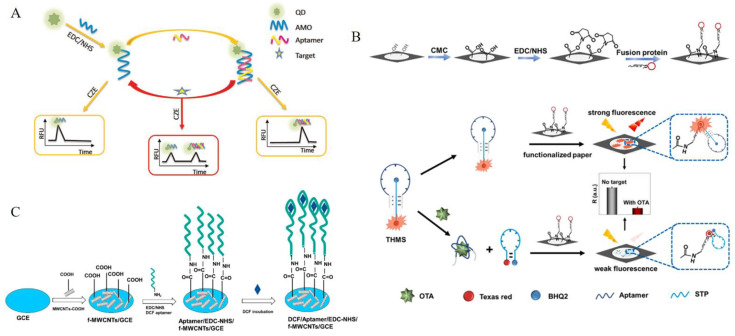
The principles of EDC/Sulfo–NHS chemical conjugations. (**A**) The detection of pesticides. Reproduced with permission from [37]; published by Elsevier, 2016. (**B**) The detection of chloramphenicol. Reproduced with permission from [38]; published by Elsevier, 2023. (**C**) The detection of diclofenac. Reproduced from [39].

**Figure 3 foods-13-01749-f003:**
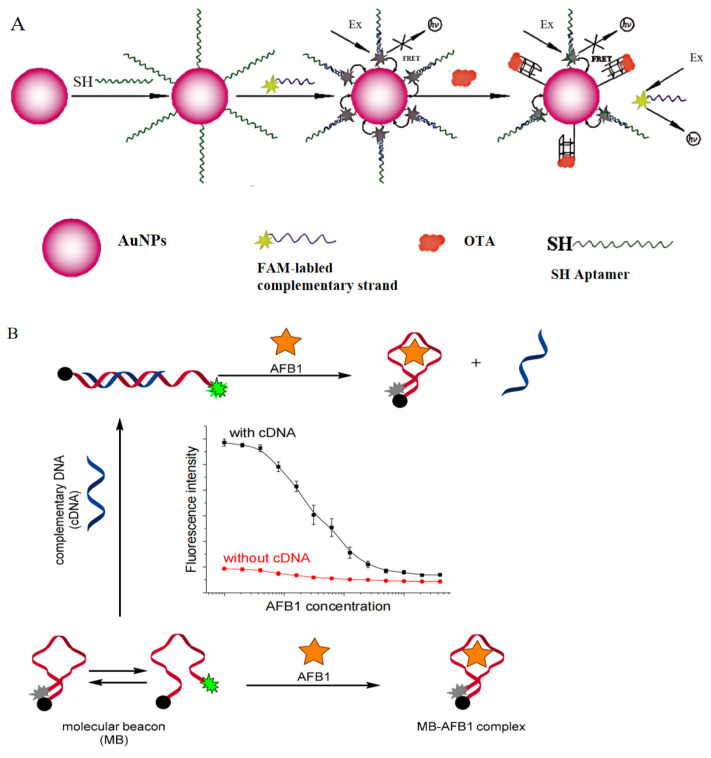
The principles of the nucleic acid hybridization method. (**A**) The detection of ochratoxin A. Reproduced with permission from [[41]#]; published by TAYLOR & FRANCIS INC, 2012. (**B**) The detection of aflatoxin B1. Reproduced from [42]. (**C**) The detection of chloramphenicol. Reproduced with permission from [43]; published by ROYAL SOC CHEMISTRY, 2015.

**Figure 5 foods-13-01749-f005:**
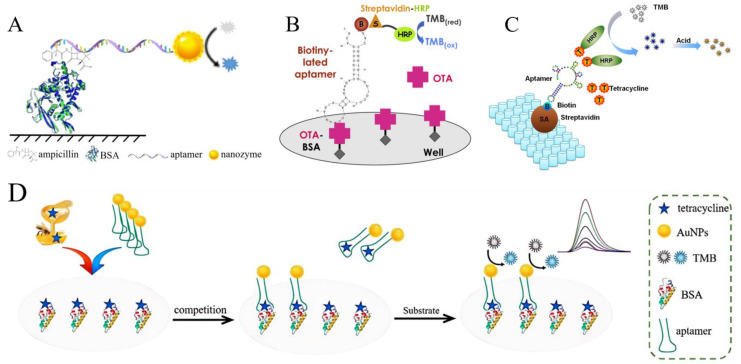
The principle of aptamer−based enzyme−linked immunity assays. (**A**) The detection of ampicillin in milk. Reproduced with permission from [73]; published by Springer, 2022. (**B**) The detection of ochratoxin A in wine. Reproduced with permission from [78]; published by Elsevier, 2011. (**C**) The detection of tetracycline in honey. Reproduced with permission from [79]; published by Elsevier, 2015. (**D**) The detection of tetracycline in honey. Reproduced from [80]. (**E**) The detection of aflatoxin B1 in peanuts. Reproduced with permission from [81]; published by Elsevier, 2020. (**F**) The detection of aflatoxin B1 in dried red chilies, groundnuts, and pepper. Reproduced with permission from [82]; published by Elsevier, 2018.

**Table 1 foods-13-01749-t001:** Comparison of the analytical capacity of ULT and other reported methods for OTA determination.

Technology	Samples	Linear Range	LOD	References
ULT	Beer	0.01~100 ng/mL	5 pg/mL	[48]
SPME−HPLC−MS	Dry red and dry white wine	−	0.02~0.05 μg/L	[50]
GC	Grain and meat	0.01~2 mg/kg	5~10 ng/mL	[51]
LFIA	Soybeans, corn, and rice	0.25~2.5 ng/mL	0.73~1.69 μg/kg	[52]
FIA	Corn	6.25~100 ng/mL	0.693 ng/mL	[53]

SPME−HPLC−MS: solid phase microextraction followed by high−performance liquid chromatography coupled with mass spectrometry; GC: gas chromatography; LFIA: lateral flow immunochromatography assay; and FIA: fluorescent immunochromatography assay.

**Table 2 foods-13-01749-t002:** Comparison of the analytical capacity of ULT and other reported methods for CBZ determination.

Technology	Samples	Linear Range	LOD	References
ULT	Apple, cucumber	0.1~5000 ng/mL	0.05 ng/mL	[56]
LFIA	Rice, soybean	5~50 ng/mL	1.6 ng/mL	[57]
EC	Milk	0.1~50 M	3.46 nM	[58]
SERS	Apple, orange, pear	0.01~1 mg/kg	9.43 μg/kg	[59]
EIMS	Apple, orange, pear, mango	0.1~5 mg/L	0.01~0.03 mg/kg	[60]

EC: electrochemical method; GC: gas chromatography; SARS: surface−enhanced Raman spectroscopy; and EIMS: electrospray ionization–ion mobility spectrometry.

**Table 3 foods-13-01749-t003:** Comparison of the analytical capacity of ULT and detection methods based on nanomaterials for OTA determination.

Technology	Nanomaterial	Samples	Linear Range	LOD	References
ULT	UCNPs	Wine	0.001~250 ng/mL	0.001 ng/mL	[62]
LFIA	MNPs	Astragalus membranaceus	0.2~20 ng/mL	0.053 ng/mL	[63]
SERS aptasensor	Ag@Au core/shell nanoparticles	Red wine	0.01~50 ng/mL	0.004 ng/mL	[64]
FIA	QDs	Cereal samples	0.001~200 μg/L	0.1 ng/L	[65]
FIA	Magnetic nanoparticles	Milk	0.1~2.5 ng/mL	0.1 ng/mL	[66]

**Table 4 foods-13-01749-t004:** The capacity of Apt−ELISA for detecting small molecules in samples.

Targets	Samples	Linear Range	LOD	References
Ampicillin	Milk	0.03~10 ng/mL	0.0013 ng/mL	[73]
Ochratoxin A	Wine	−	1 ng/mL	[78]
Tetracycline	Honey	0.1~1000 ng/mL	0.0978 ng/mL	[79]
		0.01~10 ng/mL	0.0027 ng/mL	[80]
Aflatoxin B1	Peanut	0.01~1000 ng/mL	5 pg/mL	[81]
	Dried red chilies, groundnuts, and pepper	1 pg/mL~1 ng/mL	1 pg/mL	[82]

**Table 5 foods-13-01749-t005:** The capacity of Apt−based LFDs for detecting small molecules in samples.

Technology	Targets	Samples	Linear Range	LOD	References
Nano enzyme−Apt−LFD	Acetamiprid	Tomato, rape	5~200 ng/mL	0.03 ng/mL	[85]
	Aflatoxin B1	Peanut, maize, wheat	0.01~50 ng/mL	2.2 pg/mL	[86]
AuNPs−Apt−LFD	Acetamiprid	Tomato, rape	5~200 ng/mL	0.33 ng/mL	[89]
	Kanamycin	Honey	50~1250 ng/mL	250 ng/mL	[90]
	Microcystins	Fish	1~50 ng/mL	0.84 ng/mL	[93]
	Zearalenone	Corn	5~200 ng/mL	20 ng/mL	[94]

**Table 6 foods-13-01749-t006:** The capacity of FAS for detecting small molecules in samples.

Technology	Targets	Samples	Linear Range	LOD	References
Labeled FAS	Enrofloxacin	Fish, shrimp	0.1~1000 ng/mL	0.034 ng/mL	[98]
	Zearalenone, ochratoxin A	Cornflour	0.5~100 ng/mL 0.1~50 ng/mL	0.44 ng/mL 0.098 ng/mL	[102]
Label−free FAS	Pesticide	Rice	0.3~30 nM	0.085 nM	[105]
	Aflatoxin B1	Soy sauce	5~500 ng/mL	1.83 ng/mL	[106]

**Table 7 foods-13-01749-t007:** The capacity of EASs with different labels/probes for detecting small molecules in samples.

Targets	Redox Probes	Samples	Linear Range	LOD	References
Ochratoxin A	−	Coffee beans	0.002~1 ng/mL	1 pg/mL	[108]
Acetamiprid	Glu−GQD/Au	Spinach, green beans	0.1~1 × 10^5^ fM	0.37 fM	[115]
Aflatoxin B1	Fc	Corn	0.1~10,000 pg/mL	0.012 pg/mL	[116]
Amoxicillin		Milk	100 fM~100 nM	19.57 fM	[119]
Aflatoxin B1	MB	Peanut	1 pg/mL~100 ng/mL	1 × 10^−3^ ng/mL	[120]
Bisphenol A	[Fe (CN)6]^3−/4−^	Milk, orange juice	0.1~100 nM	10 pM	[121]
Patulin	MB−Fc	Apple, pear, tomato	0.1 nM~100.0 µM	0.043 nM	[122]
Malathion, chlorpyrifos	THI−Fc	Celery, tomato, apple	1.0 μM~0.1 pM	0.038 pM, 0.045 pM	[123]
Aflatoxin B1, ochratoxin A	Hemin@HKUST−1, ferrocene@HKUST−1	Cornflour	1.0 × 10^−2^~1.0 × 10^2^ ng/mL	4.3 × 10^−3^ ng/mL, 5.2 × 10^−3^ ng/mL	[124]

Glu−GQD/Au: glutamic acid−functionalized graphene quantum dots; MB: molecular beacon; and THI−Fc: thionine and ferrocene.

**Table 8 foods-13-01749-t008:** Comparison of different CASs for aflatoxin B1 detection in samples.

Samples	Signal Probes	Linear Range	LOD	References
Cereal	AuNPs@SH−poly A	20~900 ng/mL	10 ng/mL	[126]
Rice, peanut	AuNPs	1~10 ng/mL	0.18~0.36 ng/mL	[127]
Vinegar, wine, peanut	MNPs	1~30 μg/mL	0.54 fg/mL	[128]
Oil, peanut, grain, soybean	Cy5	0.1~1000 ng/mL	0.1 ng/mL	[129]

**Table 9 foods-13-01749-t009:** Comparison between IASs and LFISs in mycotoxins.

Technology	Labels/Probes	Targets	Samples	Linear Range	LOD	References
IAS	AuNPs	Ochratoxin A	Milk, apple juice	0.05~25 ng/mL	0.05 ng/mL	[134]
	AuNPs	Aflatoxin B1	Cornflour	0.1~50 ng/mL	0.1 ng/mL	[137]
	Cy5	Type−B aflatoxins	Nuts, dried figs	0.2~20 ng/mL	0.16 ng/mL	[138]
LFIS	−	Aflatoxin B1	Guar and barley	10~100 ng/mL	3.8 ng/mL	[139]
	−	Ochratoxin A	Cereals, pork	0.37~2.86 ng/mL	0.19 ng/mL	[140]
	−	Aflatoxin B1	Mushrooms, okra	18.18~342.85 ng/mL	55.41 ng/mL	[141]

**Table 10 foods-13-01749-t010:** The capacity of Apt−NAT for detecting small molecules in samples.

Technology	Targets	Samples	Linear Range	LOD	References
Apt−qPCR	Ochratoxin A	Beer	0.039~1000 ng/mL	9 μg/mL	[142]
Apt−LAMP		Red wine	0.00005~100 nM	10 fM	[145]
Apt−HCR	Saxitoxin	Shellfish	78.13~2500 pM	42.46 pM	[148]
Apt−SDA	Patulin toxin	Apple juice, grape juice	0.001~100 ng/mL	0.042 pg/mL	[152]

## Data Availability

No new data were created or analyzed in this study. Data sharing is not applicable to this article.

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
