# Peer review of "Advances in Aptamer-Based Conjugate Recognition Techniques for the Detection of Small Molecules in Food"

_foods, 2024, doi:10.3390/foods13111749_

Round 1
Reviewer 1 Report
Comments and Suggestions for Authors
Dear Authors,
major revisions needed

Comments on the Quality of English Language
Moderate English editing
Author Response
Dear Reviewer #1:
Thank you very much for the time that you spent on reviewing our manuscript. Those comments are all valuable and helpful for revising and improving our paper. We have added all necessary information to clarify the findings and the details of figures. The detail could be reviewed in the revised manuscript with correction red marked. We have improved the quality of our article language by professional English editing service of MDPI (english-80822). All details can be seen in the revised manuscript with blue highlights.
Reviewer #1:
Herein, a review of aptasensors applications for detecting small molecules relevant to food safety issues.
The work is interesting, however, major revisions are needed.
Point 1: Improve the introduction: what are the differences between aptasensors and immunosensors (their best competitors)?
Response 1: Thanks for the comment. We have added the differences between aptasensors and immunosensors. (Please see the revised manuscript, lines 65-71, 457-461.)
Point 2: Toxins detection. Aflatoxins B1, and ochratoxin A are incredibly relevant in food safety. However, to show the robustness of aptasensors a comparison with immunosensors, especially when label-free the most innovative, should be provided. Please, insert tables reporting the performances also of some recent aflatoxins B1, and ochratoxin A biosensors (here are two examples, doi.org/10.1016/j.microc.2023.108868). Cite them.
Response 2: Thanks for the comment. According to your suggestions, we have added the tables reporting the performances of some recent aflatoxins B1, and ochratoxin A biosensors, and cited this reference. (Please see the revised manuscript, lines 487-491. Table 9. Reference [139], [140]).
Point 3: I suggest reporting a comparison with other methods in general otherwise it is not possible to appreciate how aptasensors are analytically robust.
Response 3: Thanks for the comment. According to your suggestions, we have added the content of comparison between the aptamer based methods and other methods in general. (Please see the revised manuscript, lines 198-200, 236-239, 264-267, 301-303, 328-330, 341-343, 375-377, 385-388, 429-432, 488-491, 510-513, 532-534, 544-546).
Point 4: A review should not be a simple summary of works, but a critical overview. Please, be critical and improve the manuscript with which according to you are the main advantages, challenges and the future perespectives.
Response 4: Thanks for the comment. We have added the content of main advantages, challenges and the future perespectives. (Please see the revised manuscript, lines 564-597).

Reviewer 2 Report
Comments and Suggestions for Authors
The subject addressed in the manuscript may be of interest to Foods readers. There are numerous unclear aspects that need clarification. The authors must extensively revise the manuscript, and here are some observations that may be helpful in this regard:
- There are grammatical and stylistic errors, and some sentences are hard to understand. I'll provide just two examples: "Aptamer is a short chain nucleic acid that can specifically bind to small molecule. As the detection techniques are used to identify the conjugates, its capability will be greatly enhanced." The aptamer cannot be a detection technique. "Immunoassay has the advantages of specificity and reagent consumption." How can reagent consumption be advantageous? Is something missing? There are many more sentences like this, so the authors are requested to seek assistance from a native English speaker or a specialist for a comprehensive revision of the manuscript.
- Please revise the units of measurement taken from regulations in paragraph 2 of the Introduction. It's not clear what they refer to: milligram per what? Milligrams per milliliter of what? Please revise.
- For a literature review, presenting unique figures taken from the literature is not important. In my opinion, all review figures should be transformed into comprehensive figures, comprising a compilation of several examples from the literature that address the same aptamer/analyte interaction mechanism, the same detection method or principle, or target aptasensors for the same analyte. For example, Figures 1 and 2 can be combined, and surely there are more examples of aptasensors using BAS for detection that can be included in this combined figure. In this regard, the authors are requested to revise all figures in the manuscript.
- Section 3.3.2. does not seem complete to me. Not only methylene blue is used as a label for the aptamer. More frequent examples are those with ferrocene as a label, but in the case of electrochemical aptasensors, indirect detection with a redox probe in solution is very commonly used. Such examples and approaches should be included in this section so that the reader does not get the idea that only this approach has been used and published.
- Table 1 needs to be moved from the Conclusions section, and a critical discussion of the data presented in the table needs to be introduced in the manuscript. The Conclusions section needs to be rewritten after moving the table and possibly transformed into Conclusions and Perspectives, which should also include trends observed in the literature analyzed by the authors for this review.
Comments on the Quality of English Language
There are grammatical and stylistic errors, and some sentences are hard to understand, so the authors are requested to seek assistance from a native English speaker or a specialist for a comprehensive revision of the manuscript.
Author Response
Dear Reviewer #2:
Thank you very much for the time that you spent on reviewing our manuscript. Those comments are all valuable and helpful for revising and improving our paper. We have added all necessary information to clarify the findings and the details of figures. The detail could be reviewed in the revised manuscript with correction red marked. We have improved the quality of our article language by professional English editing service of MDPI (english-80822). All details can be seen in the revised manuscript with blue highlights.
Reviewer #2:
The subject addressed in the manuscript may be of interest to Foods readers. There are numerous unclear aspects that need clarification. The authors must extensively revise the manuscript, and here are some observations that may be helpful in this regard:
Point 1: There are grammatical and stylistic errors, and some sentences are hard to understand. I'll provide just two examples: "Aptamer is a short chain nucleic acid that can specifically bind to small molecule. As the detection techniques are used to identify the conjugates, its capability will be greatly enhanced." The aptamer cannot be a detection technique. "Immunoassay has the advantages of specificity and reagent consumption." How can reagent consumption be advantageous? Is something missing? There are many more sentences like this, so the authors are requested to seek assistance from a native English speaker or a specialist for a comprehensive revision of the manuscript.
Response 1: Thanks for the comment. Some sentences have been modified to make the meaning more accurately. In addition, we have improved the quality of our article language by professional English editing service of MDPI (english-80822). All details can be seen in the revised manuscript with blue highlights.
Point 2: Please revise the units of measurement taken from regulations in paragraph 2 of the Introduction. It's not clear what they refer to: milligram per what? Milligrams per milliliter of what? Please revise.
Response 2: Thanks for the comment. According to your suggestions, we have revised the manuscript. (Please see the revised manuscript, lines 41-44.)
Point 3: For a literature review, presenting unique figures taken from the literature is not important. In my opinion, all review figures should be transformed into comprehensive figures, comprising a compilation of several examples from the literature that address the same aptamer/analyte interaction mechanism, the same detection method or principle, or target aptasensors for the same analyte. For example, Figures 1 and 2 can be combined, and surely there are more examples of aptasensors using BAS for detection that can be included in this combined figure. In this regard, the authors are requested to revise all figures in the manuscript.
Response 3: Thanks for the comment. According to your suggestions, we have revised all figures and transformed into comprehensive figures. (Please see the revised manuscript, Figure 1-11).
Point 4: Section 3.3.2. does not seem complete to me. Not only methylene blue is used as a label for the aptamer. More frequent examples are those with ferrocene as a label, but in the case of electrochemical aptasensors, indirect detection with a redox probe in solution is very commonly used. Such examples and approaches should be included in this section so that the reader does not get the idea that only this approach has been used and published.
Response 4: Thanks for the comment. According to your suggestions, we have supplemented the references of electrochemical aptasensor based on different redox probes. (Please see the revised manuscript, lines 378-399, Table 7).
Point 5: Table 1 needs to be moved from the Conclusions section, and a critical discussion of the data presented in the table needs to be introduced in the manuscript. The Conclusions section needs to be rewritten after moving the table and possibly transformed into Conclusions and Perspectives, which should also include trends observed in the literature analyzed by the authors for this review.
Response 5: Thanks for the comment. According to your suggestions, Table 1 have been moved from the conclusions section and split into corresponding parts in the manuscript. In addition, the conclusions section have been rewritten (Please see the revised manuscript, lines 563-596, Table 1-10)
Reviewer 3 Report
Comments and Suggestions for Authors
The present review work entitled "Advances in Aptamer Conjugate Recognition Technique for Small Molecules Detection in Food" presents the summary of a compilation of research works in which aptamers are used for the detection of small molecules through different techniques such as; in vitro detection using the biotin-avidin system, EDC/Sulfo-NHS chemical conjugation, nucleic acid hybridization, Upconversion luminescent nanomaterials, immunoassays, lateral flow tests, biosensors, qPCR, among others.
The work focuses on an important aspect to investigate, such as the detection of small molecules in foods, which is one of the applications and not the only one of aptamer technology. The review is well written, easy to read, and contains a good explanation of the techniques studied in each investigation with figures that clearly illustrate the techniques.
· That's where my first observation comes from. Do you have the necessary permissions to disseminate third-party images published in other scientific journals?
· The quality of the figures is not the best either and it is recommended to improve it.
· Furthermore, I consider that table 1 at the end of the writing after the conclusions is not the best place.
· The citation of figure 4 in the text is after the presentation of the figure which makes it a bit confusing, the same as figures 9 and 11.
· The selection criteria for the chosen works are not mentioned, which is of great importance for the transparency of the review. Most of the reviewed works belong to the same authors.
Author Response
Dear Reviewer #3:
Thank you very much for the time that you spent on reviewing our manuscript. Those comments are all valuable and helpful for revising and improving our paper. We have added all necessary information to clarify the findings and the details of figures. The detail could be reviewed in the revised manuscript with correction red marked. We have improved the quality of our article language by professional English editing service of MDPI (english-80822). All details can be seen in the revised manuscript with blue highlights.
Reviewer #3:
The present review work entitled "Advances in Aptamer Conjugate Recognition Technique for Small Molecules Detection in Food" presents the summary of a compilation of research works in which aptamers are used for the detection of small molecules through different techniques such as; in vitro detection using the biotin-avidin system, EDC/Sulfo-NHS chemical conjugation, nucleic acid hybridization, Upconversion luminescent nanomaterials, immunoassays, lateral flow tests, biosensors, qPCR, among others.
The work focuses on an important aspect to investigate, such as the detection of small molecules in foods, which is one of the applications and not the only one of aptamer technology. The review is well written, easy to read, and contains a good explanation of the techniques studied in each investigation with figures that clearly illustrate the techniques.
Point 1: That's where my first observation comes from. Do you have the necessary permissions to disseminate third-party images published in other scientific journals?
Response 1: Thanks for the comment. We have obtained the copyright permissions from the corresponding publisher of each reference. (Please see the following list).
Point 2: The quality of the figures is not the best either and it is recommended to improve it.
Response 2: Thanks for the comment. According to your suggestions, we have improved the quality of the figures. (Please see the revised manuscript, Figure 1-11).
Point 3: Furthermore, I consider that table 1 at the end of the writing after the conclusions is not the best place.
Response 3: Thanks for the comment. According to your suggestions, Table 1 have been moved from the conclusions section and split into corresponding parts in the manuscript. (Please see the revised manuscript, Table 1-10).
Point 4: The citation of figure 4 in the text is after the presentation of the figure which makes it a bit confusing, the same as figures 9 and 11.
Response 4: Thanks for the comment. According to your suggestions, All the positions of the figures and tables have been moved after the corresponding text. (Please see the revised manuscript).
Point 5: The selection criteria for the chosen works are not mentioned, which is of great importance for the transparency of the review. Most of the reviewed works belong to the same authors.
Response 5: Thanks for the comment. We have reclassified the figures according to methodology, while also supplemented some literature and figures. (Please see the revised manuscript, Figure 1-11)

Round 2
Reviewer 1 Report
Comments and Suggestions for Authors
Dear Authors,
the manuscript is greatly improved. Whereby, now it is ready for its accpetance
Reviewer 2 Report
Comments and Suggestions for Authors
The authors have carefully considered the suggestions and comments. The manuscript has been revised and supplemented with the requested information. The content is more logical, and the critical discussions have been enhanced. The figures are appropriately chosen. In this form, the manuscript can be recommended for publication.
Reviewer 3 Report
Comments and Suggestions for Authors
The authors have implemented the vast majority of the comments that were suggested by the reviewers.
The review article now has a more logical order, the figures have the respective permissions for their reproduction and the quality of the figures has increased significantly.
This work has interesting information that in its current state can be published and will undoubtedly be of interest among researchers working in this area.